# An Electrochemical Aptasensor Integrating Zeolitic Imidazolate Framework for Highly Selective Detection of Bioaerosols

**DOI:** 10.3390/bios12090725

**Published:** 2022-09-04

**Authors:** Pu Wang, Rui Zhang, Yunping Wu, Yangyang Chang, Meng Liu

**Affiliations:** Key Laboratory of Industrial Ecology and Environmental Engineering (Ministry of Education) and Dalian POCT Laboratory, School of Environmental Science and Technology, Dalian University of Technology, Dalian 116024, China

**Keywords:** bioaerosol, electrochemical aptasensor, ZIF-67, covalent organic frameworks

## Abstract

Bioaerosols are the biological materials in the air, which may cause a continuous threat to human health. However, there are many challenges in monitoring bioaerosols such as lack of sensitivity and selectivity. Herein, we synthesized a series of nanohybrids containing zeolitic imidazolate frameworks (ZIFs) and covalent organic frameworks (COFs) to construct an electrochemical aptasensor for detecting adenosine triphosphate (ATP), a biomarker for bioaerosols. The synthesized nanohybrids can not only improve the selectivity of aptasensor because of the original crystal and chemical features of ZIF-67, but also boost its sensitivity due to the excellent conductivity of COFs. After optimizing the nanohybrids, the novel developed sensing platform achieved highly selective detection of ATP with an excellent detection limit of 0.11 nM in a wide linear range from 0.1 nM to 100 nM. Furthermore, this assay was applied to detect bioaerosols in real air samples, and the result showed a positive correlation with that of the culturing-based method, suggesting its potential applicability.

## 1. Introduction

Bioaerosols are atmospheric particles that are derived from biological substances such as fungi, fungal spores, bacteria, viruses, pollens, or plant fibers. Bioaerosols can drift long distances and remain in the atmosphere for a long time due to their small size and lightweight, posing severe human health threats [1,2,3,4]. The World Health Organization (WHO) has given guidelines for the airborne microorganism concentration indoors which is less than 100 to 1000 CFU/m^3^ [5,6]. Therefore, monitoring bioaerosol is of great significance to protect humans against airborne biological particles. For now, the most widely used detection method for bioaerosols can be divided into culturing- and non-culturing-based methods. However, the culturing-based colony counting methods require several days for bioaerosol analysis [7]. The common non-culturing-based method such as mass spectrometry and polymerase chain reaction (PCR) suffer from several limitations such as being time-consuming, expensive, procedurally complicated, and labor-intensive [8]. Establishing a rapid and simple detection method for bioaerosols is urgent.

Sensors are a useful alternative to overcome the above limitations of culturing- and non-culturing-based methods. ATP, as the primary energy currency in cellular metabolism and cellular energy state, has been used as a biomarker for detecting total bioaerosols [9,10,11]. Although various ATP sensing methods such as fluorescence, chemiluminescence, and colorimetry have been developed, the low stability and high background signals of these methods cannot satisfy the demand for an ultrasensitive ATP determination [12,13,14,15]. The electrochemical assay, as an ultrasensitive technique, has been extensively used in the detection of ATP, due to its rapid response, excellent sensitivity, and low cost [16,17,18]. Furthermore, aptamers, a kind of single-stranded DNA or RNA oligonucleotides, are widely used as a recognition element to fabricate electrochemical biosensors [19,20,21,22]. Compared with antibodies, they are easy to synthesize, modify, and preserve for a long time [23]. However, most aptamer-based electrochemical biosensors suffer from poor selectivity, which disables biosensors to distinguish ATP from adenosine, adenosine diphosphate (ADP), or adenosine monophosphate (AMP) [24,25,26].

Metal–organic frameworks (MOFs) are extensively developed in biosensing applications due to their high porosity, specific surface area, and biocompatibility [27,28,29]. ZIF-67, a classic MOF, exhibits a specific binding effect on ATP, due to the strong adsorption of its adenine base and triphosphate at the same time [30,31]. Concurrently, the porous structure of ZIF-67 ensures its excellent absorption capacity to the aptamer. Based on these, we hypothesized that the selectivity of aptamer could be improved via its adsorption on ZIF-67.

To demonstrate this hypothesis, we constructed an aptamer-based electrical biosensing interface for total bioaerosols detection. We confirmed that ZIF-67 was introduced to improve the specificity of ATP detection, and melamine and cyanuric acid monomers (MCA) were incorporated to enhance the conductivity of the sensing interface [30,31,32]. This presented electrochemical biosensor exhibits excellent specificity for ATP, with more than five-fold signal-to-noise ratios for adenosine, ADP, and AMP. Meanwhile, the biosensor can obtain feedback in 30 s, achieving rapid detection of ATP in the bioaerosol.

## 2. Materials and Methods

### 2.1. Reagents and Materials

Melamine and Co(Ac)_2_·4H_2_O were purchased from Damao Chemical Reagent Factory (Tianjin, China). Cyanuric acid, 2-methylimidazole, Potassium chloride (KCl) yeast extract powder, peptone, and sodium chloride (NaCl) were purchased from Aladdin Biochemical Technology Co., Ltd. (Shanghai, China). Dimethyl sulfoxide (DMSO) was ordered from Tianjin Fuyu Fine Chemical Co., Ltd. (Tianjin, China). Methanol anhydrous was bought from Tianjin jindongtianzheng Precision Chemical Reagent Factory (Tianjin, China). K_3_[Fe(CN)_6_] and K_4_[Fe(CN)_6_]·H_2_O were obtained from Tianjin Ruijinte Chemical Co., Ltd. (Tianjin, China). ATP, ADP, AMP, guanosine triphosphate (GTP), cytidine triphosphate (CTP), uridine triphosphate (UTP), the sequence of ATP-targeted aptamer and mutant sequence (Appendix A) were purchased from Sangon Biotech Co., Ltd. (Shanghai, China). The *Escherichia coli* BL21 (*E. coli* BL21; CICC 24719) was purchased from the Agricultural Culture Collection of China (Beijing, China). All reagents were used without any further purification.

### 2.2. Apparatus

All the electrochemical tests were operated on a CHI650E electrochemical workstation (Chenhua Co., Ltd, Shanghai, China). The screen-printed electrode (SPE) containing a carbon working electrode, a carbon auxiliary electrode, and an Ag/AgCl reference electrode was purchased from Poten Co., Ltd. (Beijing, China). The morphologies of the synthesized materials were characterized by scanning electron microscopy (SEM) instrument (S4800, Hitachi Co., Ltd, Tokyo, Japan). X-ray diffraction spectra were performed on a SmartLab9kw (Rigaku Co., Ltd., Tokyo, Japan). X-ray photoelectron spectroscopy (XPS) was carried out on an ESCALAB instrument (Thermo Electron Co., Ltd., Waltham, MA, USA). Brunauer–Emmett–Teller (BET) was measured by Autosorb iQ (Anton-Parr, St. Louis, Missouri, USA). The bioaerosol samples were collected by the microbial aerosol concentrator (Qinyezhiyuan Co., Ltd., Beijing, China). The fluorescence was characterized by IX83 (Olympus, Tokyo, Japan). The optical density of cultured *E. coli* BL21 was carried out on METASH UV-6100 (Shanghai, China).

### 2.3. Synthesis of ZIF-67-MCA (ZM)

ZIF-67 was synthesized according to the reported work [33]. In brief, 1.494 g of Co(Ac)_2_·4H_2_O was dissolved in 30 mL of methyl alcohol (solution A), and 1.9704 g of 2-methylimidazole was dissolved in 120 mL of methyl alcohol (solution B), respectively. Solution B was dropped into solution A under continuous ultrasonication to form a homogeneous suspension. After standing for 24 h at room temperature, the precipitate was collected and washed with methyl alcohol three times, followed by drying at 60 °C for 24 h. The obtained purple powder was ZIF-67.

The MCA samples were synthesized by a coprecipitation method [34]. Typically, 3 g of melamine and 3.06 g of cyanuric acid were added into 120 mL and 30 mL of 2-methylsulfoxide, respectively. The mixed solution was kept overnight. The white precipitate was collected and washed with methyl alcohol three times, followed by drying at 60 °C for 24 h. The obtained white powder was MCA.

To improve the conductivity of the sensing interface, ZM was synthesized by adding different weight MCA into the ZIF-67 precursor using co-precipitation method. The other synthesized process of ZM was similar to that of ZIF-67. The obtained nanohybrids containing 600 mg, 1800 mg, and 3000 mg of MCA were denoted as ZIF-67-MCA_600_ (ZM-1), ZIF-67-MCA_1800_ (ZM-2), and ZIF-67-MCA_3000_ (ZM-3), respectively.

### 2.4. Preparation of Buffer, Redox Probe

An amount of 100 mL of 0.1 M phosphate buffer solution (PBS) buffer contained 2.4 g·L^−^^1^ KH_2_PO_4_, 14.2 g·L^−^^1^ Na_2_HPO_4_, 80.0 g·L^−^^1^ NaCl and 2.0 g·L^−^^1^ KCl. An amount of 100 mL of Luria Broth (LB) medium contained 0.5 g yeast extract powder, 1 g peptone, and 1 g NaCl per 100 mL. An amount of 5 mM [Fe(CN)_6_]^3−/4^^−^ and 0.1 M KCl were dissolved into the PBS buffer and used as the redox probe in cyclic voltammetry (CV) and electrochemical impedance spectroscopy (EIS) technique. Additionally, both the aptamer and the real sample were diluted by PBS buffer.

### 2.5. Fabrication of the Electrochemical Aptasensor

An amount of 5 mg of ZM powder was dispersed in 5 mL of ultrapure water and ultrasonicated for 30 min to form 1 mg·mL^−^^1^ homogeneous suspension. An amount of 10 μL of suspension was pipetted and dropped onto the working electrode and dried for 1 h. Before incubation of the aptamers, they were heated at 90 °C for 2 min and immediately cooled at 4 °C for 10 min. After activation, 10 μL of aptamer solution (1 μM) was dropped onto the working electrode for 30 min. Finally, the modified SPE was washed with ultrapure water to remove unbounded aptamers.

### 2.6. Performance of the Electrochemical Aptasensor

CV was first performed from −0.2 V to 0.8 V at the scan rate of 100 mV s^−^^1^ in 5 mM [Fe(CN)_6_]^3−/4−^ solution with 10 cycles for stabilizing system. Different concentrations of ATP (0.1 nM~100 nM) were added to the surface of electrode. An amount of 100 nM different nucleosides, including ADP, AMP, GTP, CTP, and UTP, were also added to investigate the selectivity of this sensing system. All measurements were tested under open circuit potential.

### 2.7. Real Sample Test

*E. coli* BL21 was first cultured in LB agar plate with continuous shaking at 250 rpm until the OD_600_ (optical density at 600 nm) reached 1. The suspension was split to release ATP under ultrasonic treatment (ultrasound 30 s, ice 2 min, repeated 6 times) and tested by the electrochemical workstation.

The bioaerosol samples were collected from the interior of the hall, toilet, and underground parking, respectively. The collected samples were concentrated in PBS buffer by concentrator with a speed of 120 L·min^−^^1^ for 60 min. The concentrated liquid was divided into two parts. The first part was spread onto the culture media and then hatched at 37 °C for 48 h for determining bioaerosols by culturing-based method. Additionally, the bioaerosols of the second were measured using our proposed electrochemical aptasensor. The solution containing the collected bioaerosols was first sonicated for splitting bacteria to release ATP (ultrasound 30 s, ice 2 min, repeated 6 times). Then, the obtained sonicated solution containing ATP was dropped on the SPE to completely cover three electrodes. Finally, the SPE was connected to the electrochemical workstation. All of the operations about dividing and coating plates were carried out in sterile biosafety cabinet.

## 3. Results

### 3.1. Working Principle of the Aptasensor

The working principle of this biosensor was illustrated in Figure 1a. ATP DNA aptamers were anchored on the surface of ZM via strong electrostatic interactions to fabricate an aptamer decorated ZM sensing interface. After ATP molecules were introduced, aptamers would bind with them to form the G-quadruplex structures, which would increase the steric hindrance to hinder the redox probe [Fe(CN)_6_]^3−/4−^ to reach the electrode surface, thus decreasing the electrochemical activity (Figure 1b).

The corresponding electrochemical behaviors were characterized by the EIS technique. The most adequate fit for EIS data was supported by a Randles equivalent circuit which contains solution resistance (R_s_), charge-transfer resistance (R_ct_), constant-phase element, and the Warburg impedance (W). Furthermore, the data can be fitted by Zview software to obtain the value of semicircle diameter in the Nyquist plots, denoted as R_ct_. When the ATP solution was added to the sensing surface, the R_ct_ value increased continuously due to the conformational changes of ATP and aptamer combination. The sensitivity of the biosensor was tested by measuring the ∆R_ct_ and concentration of ATP.

### 3.2. Characterization and Optimization of Sensing Materials

ZIF-67 was synthesized by mixing cobalt ion and 2-methylimidazolate anions. The hard metal ions connect with redox-inactive organic ligands, resulting in a low electrical conductivity [35,36]. Its morphology was examined by SEM (Figure 2a). The average size of the individual particles was around 1 μm. To improve its conductivity, MCA, which exhibits high electrochemical activity, was added to the precursor of ZIF-67. As shown in Figure 2a, MCA was the porous spherical material with a diameter of 4 μm. After adding MCA, the SEM image showed that ZIF-67 nanoparticles were dispersed on the MCA microparticles’ surface (Figure 2b), and the solution color changed from dark purple to light purple (Figure 2c). Additionally, the zeta potential of MCA, ZIF-67, and ZM were also determined, respectively. As expected, the combination of negatively charged MCA with the positively charged ZIF-67 can lead to the formation of ZM with less positively charged (Figure 2c). The above results demonstrated the successful formation of ZM.

To optimize the ratio of ZIF-67 and MCA, a series of ZM with different amounts of MCA adding was synthesized and the crystal structures were characterized by X-ray diffraction (XRD) (Figure 2d). The XRD pattern of the pure MCA showed two broad peaks at 10.6° and 27.8°, which are characteristic of the periodic arrays of interplanar stacking and interplanar aromatic stacking of MCA, respectively. ZIF-67 exhibited the main peaks in the range of 10° to 40°. For nanohybrids, the intensity of MCA peaks increased, and ZIF-67 peaks decreased with the MCA amount increasing, suggesting the formation of ZM with different contents of MCA. Based on this, the working electrodes were modified by various ZM to investigate their sensing performance (Figure 2e). We found the addition of MCA significantly improve the conductivity of ZM (orange column), and ZIF-67 provided more active sites for aptamer than MCA (green column). Combining the advantages of both materials, ZM nanohybrids not only have excellent conductivity, but also exhibit good adsorption capacity for aptamer. For three ZM, ZM-2 showed the strongest signal enhancement (purple column), due to its great electrochemical activity and response to ATP. Thus, ZM-2 was selected as the sensing material for aptamer immobilization. The X-ray photoelectron spectroscopy (XPS) results showed that Co2p, C1s, N1s, and O1s from ZIF-67 and MCA were observed at the same time in ZM-2 (Appendix A). The presence of N-C=O, N=C-N, and high content of nitrogen can improve the binding capacity of the aptamer. Meanwhile, the existence of Π-Π* from MCA can increase the electrochemical activity of nanocomposites. These results suggest that the combination of MCA and ZIF-67 has the potential to improve the sensitivity and selectivity of the biosensing interface. Additionally, Appendix A showed the N_2_ adsorption/desorption isotherm, and the specific surface area was 47.79 m^2^·g^−^^1^. The extremely high BET surface strongly supported the fact that the porous property.

### 3.3. Characterization of the Electrochemical Aptasensor

The sensor was fabricated by drop-casting ZIF-67-MCA and ATP DNA aptamers onto the working electrode (Figure 3a). The sensing film endowed the working electrode with a rough sensing surface after modification with ZIF-67-MCA, as shown in the SEM images in Figure 3b. Furthermore, the immobilization of aptamers was visualized by FAM-decorated ATP DNA aptamers. According to the weak fluorescence signals in Figure 3(cii), few aptamers were adsorbed on the bare electrode due to the lack of adsorbed sites. Obviously, Figure 3(ciii) showed a stronger fluorescence signal on the surface after ZM immobilization, which proves the successful fabrication of the sensing layer. Furthermore, Figure 3(civ) showed that the binding of the target ATP decreased the fluorescence signal, maybe due to the change of aptamer structure to form G-quadruplex, suggesting the specific binding of ATP with the nanohybrid layer.

### 3.4. Detection of ATP

We evaluated the sensing performance of the aptasensor by adding various concentrations of ATP. With increasing ATP concentration, the diameter of the semicircle Nyquist plot was enlarged (Figure 4a). Correspondingly, the ∆R_ct_ (∆R_ct_ represents the R_ct_ value of the sample with ATP adding by subtracting the R_ct_ value of the sample without ATP adding) was gradually increased. The calibration graph showed a positive collinear relationship between the ∆R_ct_ value and the log10 concentration of ATP, and the regression equation was fitted as ∆R_ct_ = 371.32 LogC_ATP_ + 479.70 (R^2^ = 0.995) with a concentration range of 0.1 nM–100 nM (Figure 4b). The limit of detection (LOD) was calculated to be 0.11 nM based on S/N = 3 (S is signal, N is noise). Compared with the previously reported aptasensors (Appendix A) [37,38,39,40,41,42], the ZM-2-based aptasensor has a lower LOD and a wider range of detection.

In addition to the high sensitivity, the developed electrochemical biosensor also exhibited excellent selectivity for ATP (Figure 4c). The selectivity coefficient is defined as the ratio of response to ATP to interferents. The selectivity coefficients of ADP, AMP, GTP, CTP, and UTP were 5.12, 4.67, 27.09, 26.57, and 21.96, respectively. Furthermore, we also investigated the possibility of the sensor for ATP detection in real-time (Figure 4d). The sensor can rapidly respond to ATP within 10 s, suggesting that it is promising for the real-time detection of ATP. In addition, we verified the repeatability, reproducibility, and stability of the aptasensor. The multi-scan cyclic voltammograms of the aptasensor were shown in Appendix A. It is noteworthy that no significant change in the peak current, suggesting good repeatability of this assay. Additionally, the relative standard deviation for six successive aptasensors was 4.53% (Appendix A), indicating its great reproducibility. Moreover, the ∆R_ct_ of the aptasensor in response to 100 nM ATP still maintained 93.7% of its initial level after 16 days, further confirming excellent stability.

### 3.5. Analysis of ATP in E. coli Cells and Real Air Samples

Before investigating the sensing performance of real bioaerosols, we first tested the lysed cultured *E. coli* suspension. As shown in Figure 5a, the released ATP concentration was calculated by the regression equation to be 0.72 nM, which is the same order of magnitude as previously reported [43]. Moreover, we changed the aptamer to a mutant sequence. The R_ct_ of the sensor only changed from 634.5 Ω to 656.8 Ω (Figure 5b) when ATP was added. These results indicate that the sensor is stimulated by the ATP released from *E. coli* to induce the electrical signal change.

To investigate the application of the aptasensor in real bioaerosols, we collected the air samples in the hall (day and night), garage, and toilet into PBS buffer by the bioaerosol concentrator (Appendix A and Figure 5c). The collected solution was divided into two parts. One part was coated on the plates to count the number of bacterial colonies. The other part was ultrasonicated to release ATP and followed by determination using our developed assay. The concentration of ATP was calculated to be 1.65 nM, 1.59 nM, 1.42 nM, and 0.91 nM based on the fitting curve (Figure 5d). Additionally, the corresponding numbers of bacteria were counted as 158 CFU/m^3^, 152 CFU/m^3^, 136 CFU/m^3,^ and 87 CFU/m^3^, respectively (Appendix A). The calculated CFU values are in the range of standard for bioaerosols suggested by the World Health Organization guidelines [44]. A positive correlation was observed between the measured ATP concentration and the counted bacterial number, suggesting the potential applicability of this aptasensor.

## 4. Conclusions

In summary, we constructed an aptasensor for bioaerosols’ detection by assembling ZM-Apt on SPE as the sensing interface. The specific combination between ATP and its aptamer leads to conformational changes and prevents electron transfer, which results in strong electrochemical impedance changes to produce the electrochemical signals. Since the ZM nanohybrids can not only provide many active sites for aptamer adsorption but also show high adsorption for ATP, this aptasensor exhibits high sensitivity and excellent selectivity. On this basis, the sensing signal showed a linear relationship with the ATP concentration, and the LOD was estimated to be 0.11 nM. Additionally, this assay can be used in bioaerosol detection and have acceptable applicability in real samples. The presented work can provide new insight into bioaerosol detection by an integrated and portable screen-printed platform and will find its useful application in environmental monitoring.

## Figures and Tables

**Figure 1 biosensors-12-00725-f001:**
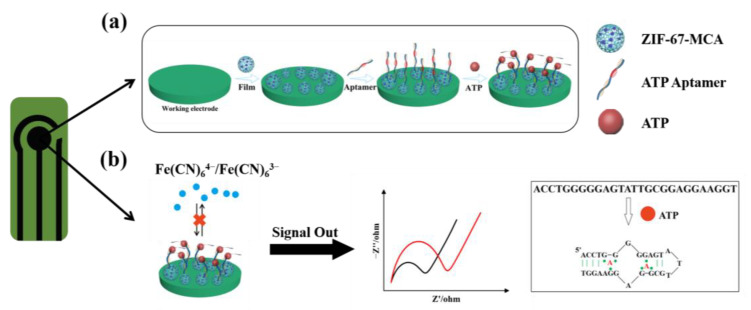
(**a**) Scheme of the process for the fabrication of ZM-based electrochemical aptasensor for detecting ATP; (**b**) principle of EIS measurement for detection and scheme of the conformational change by ATP binding with aptamer (the black line represents the initial R_ct_ of the aptasensor_,_ the red line represents the R_ct_ of the aptasensor after the addition of ATP).

**Figure 2 biosensors-12-00725-f002:**
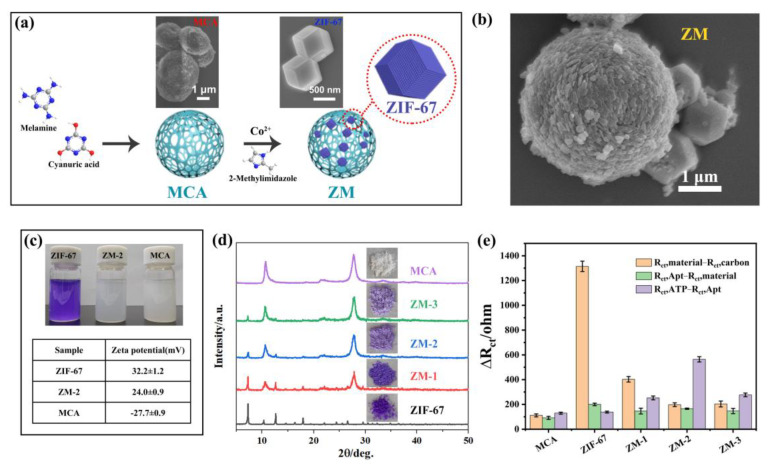
(**a**) Scheme of the synthesis of MCA, ZIF-67, and ZM. Insets are SEM images of MCA and ZIF-67; (**b**) SEM image of the ZM; (**c**) The typical photograph and zeta potential of ZIF-67, MCA, and ZM homogeneous suspension; (**d**) XRD patterns of MCA, ZIF-67, different ZM and their morphologies of powder; (**e**) the R_ct_ value variations of every step of aptasensor fabrication and ATP detection for five materials.

**Figure 3 biosensors-12-00725-f003:**
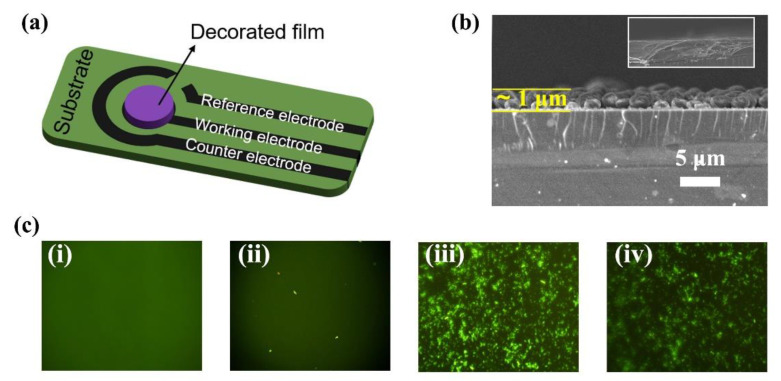
(**a**) Schematic diagram of the structure of modified carbon screen-printed electrode; (**b**) SEM images of the intersecting surface; (**c**) fluorescence images of (**i**) bare carbon working electrode, (**ii**) bare carbon working electrode/FAM aptamer, (**iii**) ZM modified carbon working electrode/FAM aptamer and (**iv**) ZM modified carbon working electrode/FAM aptamer/ATP.

**Figure 4 biosensors-12-00725-f004:**
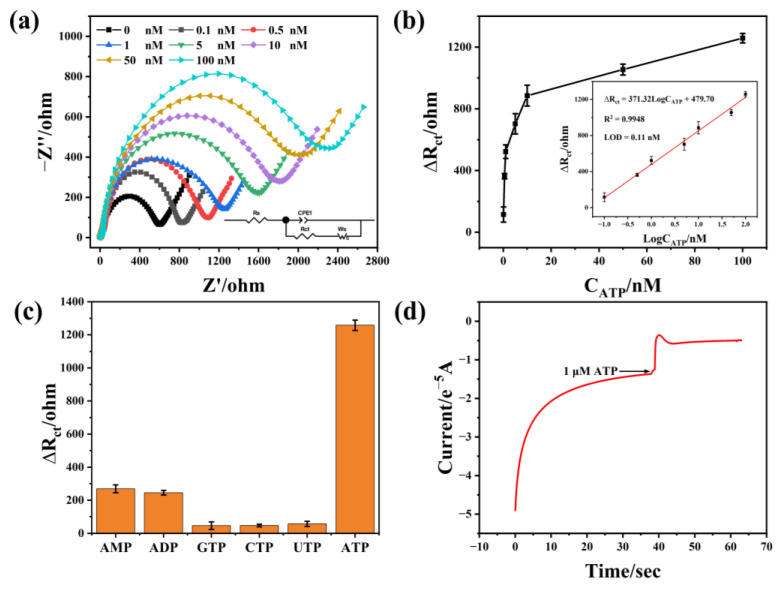
(**a**) EIS Nyquist plots for the detection of different concentrations of ATP (0, 0.1, 0.5, 1, 5, 10, 50 and 100 nM) (inset: the equivalent circuit model for EIS fitting); (**b**) the corresponding calibration curves between ΔR_ct_ and ATP concentrations (Inset: the linear calibration curve for ATP determination); (**c**) selectivity study of the aptasensor; (**d**) I–T curves or kinetics test.

**Figure 5 biosensors-12-00725-f005:**
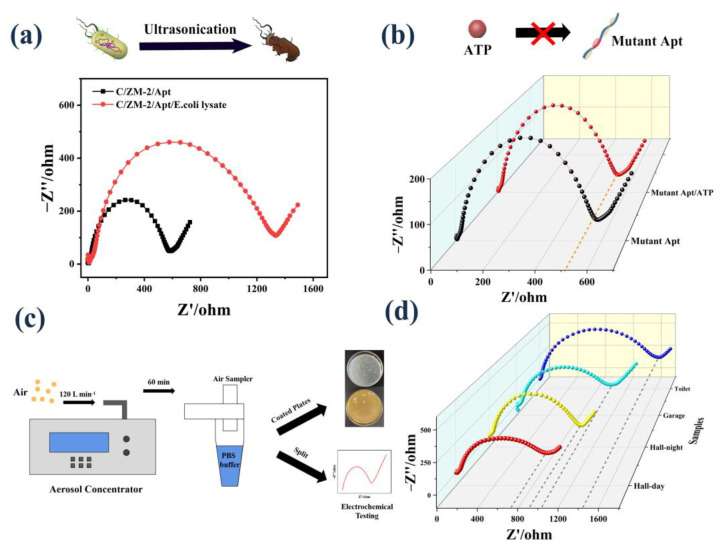
(**a**) EIS response to ATP extraction from *E. coli* cells; (**b**) EIS response of the mutant sequence modified electrode to ATP; (**c**) schematic image of bioaerosol collection, coating plates, and electrochemical test; (**d**) EIS response to the ATP released by the real samples.

## Data Availability

Not applicable.

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
