# Peer review of "An Electrochemical Aptasensor Integrating Zeolitic Imidazolate Framework for Highly Selective Detection of Bioaerosols"

_biosensors, 2022, doi:10.3390/bios12090725_

Round 1
Reviewer 1 Report
The article investigated: An Electrochemical Aptasensor Integrating Zeolitic Imidazo late Framework for Highly Selective Detection of Bioaerosols. I can recommend it for publication, as the novelty is acceptable, and also one preliminary commercial model is fabricated. Also, an excellent detection limit - 0.11 nM- is obtained, and it has potential applicability. please see my comments.
1. line, 43, “due to its low background signal, high bio-compatibility and excellent stability” is it not clear? if you talking about: electrochemical assay. I highly recommend you to modify this part, generally, the electrochemical assay is not bio-…, or low background depends on many factors!
2. line 52, MOFs, which exhibits a high selectivity for recognizing ATP., selectivity!? The aptamer was applied to provide the selectivity? how the MOFs can do it? if yes, why did you apply aptamer?
Also, you in line 61, mentioned the synthesized porous ZIF-67-MCA hybrids provide sufficient active sites for the immobilization of ATP DNA aptamer.
3. about the stability, after modification the SPE, how long the fabricated biosensor can be used? How did you keep it? did you check this factor?
4. What is lysis buffer? Add more information.
5. the “Conclusions’ part is weak. It does not show what you got. Please improve it.
6. please add these related references, aptamer, ATP detection, etc.
--Seok, Youngung, Joonseok Lee, and Min-Gon Kim. "Based Airborne Bacteria Collection and DNA Extraction Kit." Biosensors 11.10 (2021): 375.
--- Ali, A. A., Altemimi, A. B., Alhelfi, N., & Ibrahim, S. A. (2020). Application of biosensors for detection of pathogenic food bacteria: a review. Biosensors, 10(6), 58.
---- Suo, Yuanjie, Weihong Yin, Qiangyuan Zhu, Wenshuai Wu, Wenjian Cao, and Ying Mu. "A Specific and Sensitive Aptamer-Based Digital PCR Chip for Salmonella typhimurium Detection." Biosensors 12, no. 7 (2022): 458.
Author Response
All the authors would like to express our gratitude to the referees for their constructive comments. Our response to each comment is provided below following each quoted comment.
Referee 1:
<Comments>
The article investigated: An Electrochemical Aptasensor Integrating Zeolitic Imidazo late Framework for Highly Selective Detection of Bioaerosols. I can recommend it for publication, as the novelty is acceptable, and also one preliminary commercial model is fabricated. Also, an excellent detection limit - 0.11 nM- is obtained, and it has potential applicability. please see my comments.
1.line, 43, “due to its low background signal, high bio-compatibility and excellent stability” is it not clear? if you talking about: electrochemical assay. I highly recommend you modify this part, generally, the electrochemical assay is not bio-…, or low background depends on many factors!
Response: We have revised it in the revised manuscript, i.e., “The electrochemical assay, as an ultrasensitive technique, has been extensively used in the detection of ATP, due to its rapid response, excellent sensitivity, and low cost.”
- line 52, MOFs, which exhibits a high selectivity for recognizing ATP., selectivity!? The aptamer was applied to provide the selectivity? how the MOFs can do it? if yes, why did you apply aptamer? Also, you in line 61, mentioned the synthesized porous ZIF-67-MCA hybrids provide sufficient active sites for the immobilization of ATP DNA aptamer.
Response: In this work, the electrochemical signal change was mainly induced by the conformational change of aptamer in the presence of ATP. Thus, aptamer was used as the recognition element for ATP detection. However, the selectivity of the aptamer is poor, which can not distinguish ATP from adenosine, adenosine diphosphate, or adenosine monophosphate. Liu et al reported that ZIF-67 has excellent selectivity for ATP, because of the strong adsorption of its tri-phosphate and phosphate backbone at the same time. Concurrently, the porous ZIF-67 can provide sufficient active sites for the immobilization of aptamer. Based on these, the selectivity of aptamer for ATP detection was improved via its adsorption on ZIF-67. Correspondingly, we have added the sentences, i.e., “ZIF-67, a classic MOF, exhibits a specific binding effect to ATP, due to the strong adsorption of its adenine base and triphosphate at the same time.[31,32] Concurrently, the porous structure of ZIF-67 ensures its excellent absorption capacity to aptamer. Based on these, we hypothesized that the selectivity of aptamer could be improved via its adsorption on ZIF-67.” in the revised manuscript.
- about the stability, after modification the SPE, how long the fabricated biosensor can be used? How did you keep it? did you check this factor?
Response: After modification of the SPE, we kept them at 4 °C until the test. We have added a related test for determining the stability of the modified SPE in part 3.4. And the corresponding result, i.e., “Additionally, the ∆Rct of the aptasensor in response to 100 nM ATP still maintained 93.7% of its initial level after 16 days, further confirming excellent stability.” has been added in the revised manuscript. Concurrently, Fig. S5c has been added in the revised SI as follows.
Fig. S5. (c) stability of the ZM-2-based aptasensor for detection of 100 nM ATP.
- What is lysis buffer? Add more information.
Response: To avoid misunderstanding, we have revised “lysis buffer” to “sonicated solution” in the revised manuscript (Part 2.7). We have added the sentence, i.e., “and then the sonicated solution was dropped on the SPE to completely cover three electrodes.” in the revised manuscript.
- the “Conclusions’ part is weak. It does not show what you got. Please improve it.
Response: We have modified it in the revised manuscript, i.e., “In summary, we constructed an aptasensor for bioaerosols’ detection by assembling ZM-Apt on SPE as the sensing interface. The specific combination between ATP and its aptamer leads to conformational changes and prevents electron transfer, which results in strong electrochemical impedance changes to produce the electrochemical signals. Since the ZM nanohybrids can not only provide many active sites for aptamer adsorption, but also show high adsorption for ATP, this aptasensor exhibits high sensitivity and excellent selectivity. On this basis, the sensing signal showed a linear relationship with the ATP concentration, and the LOD was estimated to be 0.11 nM. Additionally, this assay can be used in bioaerosol detection and have acceptable applicability in real samples. The presented work can provide new insight into bioaerosol detection by an integrated and portable screen-printed platform, and will find its useful application in environmental monitoring.”
- please add these related references, aptamer, ATP detection, etc.
--Seok, Youngung, Joonseok Lee, and Min-Gon Kim. "Based Airborne Bacteria Collection and DNA Extraction Kit." Biosensors 11.10 (2021): 375.
--- Ali, A. A., Altemimi, A. B., Alhelfi, N., & Ibrahim, S. A. (2020). Application of biosensors for detection of pathogenic food bacteria: a review. Biosensors, 10(6), 58.
---- Suo, Yuanjie, Weihong Yin, Qiangyuan Zhu, Wenshuai Wu, Wenjian Cao, and Ying Mu. "A Specific and Sensitive Aptamer-Based Digital PCR Chip for Salmonella typhimurium Detection." Biosensors 12, no. 7 (2022): 458.
Response: We have added the related references in the revised manuscript.
Reviewer 2 Report
The authors developed an efficient electrochemical aptasensor integrating zeolitic Imidazolate framework for highly selective detection of bioaerosols. Although being interesting and informative, I find that there are some major issues with the paper that require addressing prior to this being considered for publication in this journal. I have identified the main points for consideration below:
1.This manuscript has some spelling typos, style errors and grammatical errors. Pleases carefully check the manuscript thoroughly.
2. The selectivity coefficients for all interferences should be also determined.
3. The equivalent circuit model for EIS fitting should be given in the revised manuscript.
4.The repeatability, reproducibility, and stability of the proposed electrochemical aptasensor should be added in the revised manuscript.
5. The sensing properties such as linear response range, limit of detection (LOD) for detecting ATP should be compared to the previously reported ones.
6. The full names of acronyms should be given when first mentioned.
7. Some related references related to electrochemical sensors are recommended to be cited, such as Journal of Hazardous Materials 436 (2022) 129107, Microchemical Journal 179 (2022) 107515, Sensors and Actuators B: Chemical, 2022: 132318.
Author Response
All the authors would like to express our gratitude to the referees for their constructive comments. Our response to each comment is provided below following each quoted comment.
Referee 2:
<Comments>
The authors developed an efficient electrochemical aptasensor integrating zeolitic Imidazolate framework for highly selective detection of bioaerosols. Although being interesting and informative, I find that there are some major issues with the paper that require addressing prior to this being considered for publication in this journal. I have identified the main points for consideration below:
- This manuscript has some spelling typos, style errors and grammatical errors. Pleases carefully check the manuscript thoroughly.
Response: We have carefully checked and corrected them in the revised manuscript.
- The selectivity coefficients for all interferences should be also determined.
Response: We have added the sentence, i.e., “The selectivity coefficient is defined as the ratio of response to ATP to interferents. The selectivity coefficients of ADP, AMP, GTP, CTP, UTP were 5.12, 4.67, 27.09, 26.57 ,21.96, respectively.” in the revised manuscript. (Part 3.4).
- The equivalent circuit model for EIS fitting should be given in the revised manuscript.
Response: We have added the equivalent circuit model for EIS fitting (Fig. 4a inset).
Fig. 4a (Inset: The equivalent circuit model for EIS fitting)
- The repeatability, reproducibility, and stability of the proposed electrochemical aptasensor should be added in the revised manuscript.
Response: We have added the repeatability, reproducibility and stability in the revised manuscript (Part 3.4). And we have added the corresponding sentence i.e., “In addition, we verified the repeatability, reproducibility, and stability of the aptasensor. The multi-scan cyclic voltammograms of the aptasensor were shown in Fig. S5a. It is noteworthy that no significant change in the peak current, suggesting good repeatability of this assay. And the relative standard deviation for six successive aptasensors was 4.53% (Fig. S5b), indicating its great reproducibility. Moreover, the ∆Rct of the aptasensor in response to 100 nM ATP still maintained 93.7% of its initial level after 16 days, further confirming excellent stability.” Concurrently, Fig. S5 has been added in the revised SI as follows.
Fig. S5. (a) The repeatability, (b) reproducibility, and (c) stability of the ZM-2-based aptasensor for detection of 100 nM ATP.
- The sensing properties such as linear response range, limit of detection (LOD) for detecting ATP should be compared to the previously reported ones.
Response: We have added Table S2 to compare these sensing properties in the revised Supplementary. Correspondingly, we have added the sentences, i.e., “Compared with the previous reported aptasensors (Table S2), ZM-2-based aptasensor has a lower LOD and wider range of detection.” in the revised manuscript
Supplementary Table S2. Various detection methods for ATP in some reported works.
|
Detection method |
Detection range |
LOD |
Ref |
|
EC/MOF |
100 nm-1000 μM |
3.9 nM |
[1] |
|
RLS/aptasensor |
2.5-75 nM |
0.046 nM |
[2] |
|
ECL/QDs |
8-2000 nM |
7.6 nM |
[3] |
|
CD/aptasensor |
1.5-4.2 mM |
0.2 mM |
[4] |
|
FL/GO |
0.5-250 μM |
100 nM |
[5] |
|
EC/exonuclease |
0.1-20 nM |
34 pM |
[6] |
|
ZIF-67-MCA/aptasensor |
0.1-100 nM |
0.11 nM |
this work |
- The full names of acronyms should be given when first mentioned.
Response: We have added it in the revised manuscript.
- Some related references related to electrochemical sensors are recommended to be cited, such as Journal of Hazardous Materials 436 (2022) 129107, Microchemical Journal 179 (2022) 107515, Sensors and Actuators B: Chemical, 2022: 132318.
Response: We have added the related references in the revised manuscript.
Round 2
Reviewer 1 Report
Thanks for your answers
please, in line 263, "lysed cultured " as you said in your answers, "lysis" is changed to "sonicated solution ", about line 144, please explain more, sonicated! you sonicated what ?!
please answer and modify this Q. after that, it can be accepted!
Author Response
All the authors would like to express our gratitude to the referees for their constructive comments. Our response to each comment is provided below following each quoted comment.
Referee 1:
<Comments>
1.please, in line 263, "lysed cultured " as you said in your answers, "lysis" is changed to "sonicated solution ", about line 144, please explain more, sonicated! you sonicated what ?!
Response: According to the reviewer’s suggestion, we added more explanation for “sonicated solution”. We sonicated the solution containing the collected bioaerosols, aiming to split the bacteria in bioaerosols to release ATP that can be detected by the aptasensor. We have modified it in the revised manuscript (Part 2.7), i.e., “The solution containing the collected bioaerosols was first sonicated for splitting bacteria to release ATP (ultrasound 30 s, ice 2 min, repeated 6 times). And then the obtained sonicated solution containing ATP was dropped on the SPE to completely cover three electrodes.”
Reviewer 2 Report
The authors have addressed all my questions, and I think this manuscript is now acceptable for publication.
Author Response
Thanks for your approval.